# Investigation of β-Carboline Alkaloid Harmaline Against *Cyvirus cyprinidallo3* Infection In Vitro and In Vivo

**DOI:** 10.3390/v17050687

**Published:** 2025-05-09

**Authors:** Clement Manes, Kristen Larson, Shelby Matsuoka, Xisheng Wang, Ruth Milston-Clements, Ling Jin

**Affiliations:** 1Department of Biomedical Sciences, College of Veterinary Medicine, Oregon State University, Corvallis, OR 97331, USA; clement.manes_21@envt.fr (C.M.); larsonk3@oregonstate.edu (K.L.); matsuosh@oregonstate.edu (S.M.); xisheng.wang@oregonstate.edu (X.W.); 2Department of Microbiology, College of Science, Oregon State University, Corvallis, OR 97331, USA; ruth.milston-clements@oregonstate.edu

**Keywords:** CyHV-3, temperature stress, harmaline, acyclovir, reactivation

## Abstract

*Cyvirus cyprinidallo3*, also known as *Cyprinid herpesvirus 3* (CyHV-3), is a common pathogen of koi and common carp (*Cyprinus carpio*). Infection of CyHV-3 can lead to high mortality in fry under 4 months of age. CyHV-3 can become latent in recovered fish, and latent CyHV-3 can reactivate under stress conditions and spread the virus. Reactivation of CyHV-3 can also lead to mortality and diseases in latently infected fish. No effective drugs are available to prevent CyHV-3 infection or reactivation from latency. There is a need for the discovery of anti-CyHV-3 drugs. Harmine (HAR) and harmaline (HAL) are β-carboline alkaloids found in the medicinal plant *Peganum harmala* with antiviral activities against many viruses, including HSV. Here, HAL was evaluated against CyHV-3 infection in vitro and in vivo, respectively. Immediately after a one-hour infection exposure of ~1000 FPU/plate or ~500 PFU/plate, cells treated with 5 µM HAL for 2 h can block nearly 50% or 90% plaque formation in vitro. Only around 50% inhibition was observed in cells treated with the common anti-herpesvirus drug acyclovir (ACV) at 10 or 20 µM for 2 h following 1 h post-infection of ~500 PFU/plate. Cells treated with 10 µM HAL for 30 min, 60 min, 2 h, and 6 h can reduce 60%, 65%, 85.5%, and 85% CyHV-3 replication in vitro, respectively. HAL at 20 µM is still effective against CyHV-3 DNA replication and virion production when the treatment started at 3 and 5 days post-infection for 1 or 2 h, respectively. HAL under 50 µM has little toxicity to cells treated for 24 h. Immersion treatment with 10 µM HAL for 3–4 h daily within the first 5 days post-infection can increase the survival of fry by 60%. In addition, IM injection of HAL at 20 µM can reduce the rate of CyHV-3 reactivation induced by heat stress in latently infected koi. This study demonstrated that HAL could potentially be used to prevent CyHV-3 infection or reactivation from latency.

## 1. Introduction

*Cyvirus cyprinidallo3*, also known as *Cyprinid herpesvirus 3* (CyHV-3), or commonly known as koi herpesvirus, is a pathogen of koi and common carp (*Cyprinus carpio*) that causes significant disease and mortality in young fry under 4 months of age [1,2]. The mortality of juveniles from CyHV-3 infection can reach 80–100% [2,3]. The acute CyHV-3 infection can cause necrosis and hemorrhage of the gills (red and white mottled gills), sunken eyes, pale patches or blisters on the skin, and external hemorrhages [1,2,3]. The virus can also infect the kidney, spleen, fin, intestine, and brain [4,5]. CyHV-3 is a member of the family *Alloherpesviridae* within the order of *Herpesvirales* [6]. Like other herpesviruses from *Herpesviridae*, CyHV-3 can also become latent in recovered fish from either symptomatic or asymptomatic infections [7,8,9]. Therefore, healthy-looking koi or common carp can harbor CyHV-3 latent infection without clinical signs [9,10]. CyHV-3 reactivation from latency under stress conditions can cause inflammation in various organs and death, and infectious viruses from reactivation can spread to naïve koi and cause disease [11]. Currently, no effective treatment has been developed to control CyHV-3 infection in koi, one of the most popular ornamental fish. There is a need for anti-CyHV-3 drugs to be developed to protect CyHV-3 infection in koi.

Harmaline (HAL) and harmine (HAR) are pharmacological β-carboline (βC) alkaloids with similar chemical structures that are widely distributed in *Peganum harmala* plants [12]. They possess significant inhibitory activities against acetylcholinesterase (AChE), monoamine oxidase (MAO), and myeloperoxidase (MPO) [13,14] They are found to be potent antioxidants and have anti-inflammatory, antitumor, and anti-hypertensive effects [13]. Both natural and synthetic βC alkaloids were shown to have inhibitory effects against cancerous cells, parasites, fungi, bacteria, and viruses [15]. HSV-1 was the first virus reported to be affected by certain βC alkaloids (Eudistomins) obtained from the colonial tunicate Eudistoma olivaceum [16]. Since then, the 9-butyl-harmol, a βC derivative, was demonstrated to be effective against paramyxoviruses [17] and 9N-methylamine, another synthetic βC derivative, was found to be effective against DENV-2 infection [18]. HSV infection was shown to be affected by HAR and the 9,9′-norharmane dimers (nHo-dimer), which also belongs to the βC group [15,19,20]. It has been shown that HSV-2 infection can be blocked by HAR in vitro, with an EC50 value at around 1.47 μM and a CC50 value at around 337.10 μM [19], which suggests that HAR has high potency against herpesviruses. Since HAR is potent against MAO and MPO [14], it is speculated HAR may block HSV-induced reactive oxygen species (ROS) production via enhancing innate immunity against virus infection. Indeed, HAR was shown to able to reduce NF-κB activation, IκB-α degradation, and p65 nuclear translocation during HSV-2 infection [19]. Therefore, HAR may have a broad anti-viral effect via boosting innate immune responses against viral infections [21]. CyHV-3 is also a herpesvirus with similar replication strategies like HSV-1 and HSV-2. We hypothesized that CyHV-3 is also sensitive to βC alkaloids during replication. HAL has a similar structure to HAR, but with better solubility than HAR, and it has not been tested against CyHV-3 infection. Therefore, in this study, HAL was evaluated against CyHV-3 infection in vitro and in vivo, respectively.

## 2. Materials and Methods

### 2.1. Viruses, Cells, and Chemicals

The United States strain of CyHV-3 (CyHV-3-U) was a gift from Dr. Ronald Hedrick. The koi fin (KF-1) cell line (also a gift of Dr. Ronald Hedrick) was cultured in Dulbecco’s modified Eagle’s medium (DMEM) (Invitrogen, Carlsbad, CA, USA) supplemented with 10% fetal bovine serum (FBS) (Gemini Bio-Products, West Sacramento, CA, USA), penicillin (100 U/mL), and streptomycin (100 µg/mL) (Sigma-Aldrich, Inc., St. Louis, MO, USA) and incubated at 22 °C. CyHV-3 viral stocks were prepared by infecting KF-1 cells in 75 cm^2^ flasks with CyHV-3-U at 0.1 multiplicity of infection (MOI) and maintained in DMEM supplemented with 5% FBS, penicillin (100 U/mL), and streptomycin (100 µg/mL) and incubated at 22 °C. As reported previously, the viral stock was titrated by a standard plaque assay [8,22]. Harmaline (cat no. 51330-1G, Sigma-Aldrich) and acyclovir (ACV) were purchased from BioVision (Milptitas, CA, USA).

### 2.2. CyHV-3 Plaque Reduction Assay Following Drug Treatment

KF-1 cells were seeded in 12.5 cm^2^ tissue culture plates with approximately 2 × 10^5^ cells/plate on the day before infection. Before the treatment, three replicate plates were infected with CyHV-3 at ~1000 or ~300–500 plaque-forming units (PFU)/plate. Following a 1 h viral absorption, the cells were cultured in the presence or absence of harmaline (HAL) or ACV for different treatment times at different times post-infection and then washed once with phosphate-buffered saline (PBS) and cultured in 3 mL of 3% methylcellulose overlay media for 10 days in a 22 °C incubator. The plates were then fixed in 20% methanol and stained with 1% crystal violet, and the plaques were counted after the plates were washed with water [8,22].

### 2.3. Harmaline Cytotoxicity Assay In Vitro

Ninety-six-well plates were seeded with ~1 × 10^4^ KF-1 cells per well and grown overnight at 22 °C. The cells were washed once with PBS and treated in 100 μL media containing HAL or ACV at the indicated concentrations at 22 °C for 24 h. The drug-treated cells were then washed once with PBS and replenished with fresh DMEM containing 5% FBS and antibiotics (as described above) and further incubated for 24 h. After incubation, cell viability was examined by using colorimetric cell viability kit III (XTT) (PromoKine, Huissen, The Netherlands). Briefly, 50 μL of the XXT reaction solution was added to each well and then the plate was incubated at 37 °C incubator for 3–4 h following the recommended protocols. Absorbance at a wavelength of 450 nm was read on a microplate fluoresce reader (BioTek, Winoosk, VT, USA) and recorded. Wells containing harmaline or ACV at each concentration in media without cells served as a blank to ensure that the drugs themselves were not registering fluorescence.

### 2.4. Koi Infection and Drug Treatment

Koi at 5–8 cm (3–4 months old), including both males and females, were acquired from a local distributor in Oregon, quarantined for 10 days, and tested free of CyHV-3 before infection. CyHV-3 infection was carried out by immersing koi in 1× PBS containing ~1 × 10^3^ PFU/mL of CyHV-3 for 30 min and then returning the CyHV-3 exposed koi to tanks with one-way flow-through water and air. All treatments were given via immersion in water containing 20 µM ACV, 10 µM HAL, 20 µM HAL, or dilution media DMSO as the vehicle control for 3–4 h on days 1–5 post-infection. Additional koi at similar ages served as uninfected controls. The koi were infected under stress temperature conditions as reported previously [8,23]. All koi were kept in 4-foot diameter tanks at the Oregon State University John L. Fryer Aquatic Animal Health Lab following Institutional Animal Care and Use Committee guidelines (IACUC-2021-0194). The koi were euthanized with 250 mg/L (PPM) MS-222 at the end of the studies.

### 2.5. CyHV-3 Reactivation and Harmaline Treatment in CyHV-3 Latently Infected Koi

Koi at 25–30 cm (4–6-year-old) were recovered koi that were infected with CyHV-3 approximately 3–5 years ago and were all tested positive for CyHV-3 latency. CyHV-3-positive koi were stressed as reported previously [8]. Koi were given 100 µL of 20 µM harmaline or 100 µL of PBS daily for 5 days immediately after temperature increase. Four CyHV-3-positive koi were used as the control and were not subjected to temperature stress or treatment. Blood samples from treated or untreated control groups were taken on days 1, 3, 6, and 9 after temperature stress.

### 2.6. Total White Blood Cell (WBC) Isolation and Total DNA Extraction

WBCs were isolated from approximately 2 mL of whole blood from each koi by Ficoll-Paque PLUS (GE Healthcare), as described previously [24]. Total DNA of CyHV-3-infected KF-1 cells and WBCs were isolated by an EZNA Tissue DNA extraction kit (Omega Bio-tec, Norcross, GA, USA), and the total DNA was eluted in 75 μL of TE buffer (10 mM Tris–HCL [pH 8.0]–1 mM EDTA). The CyHV-3 genome within CyHV-3-infected KF-1 cells and WBCs was estimated by real-time PCR using 5 μL of the DNA extract, as reported previously [8].

### 2.7. CyHV-3 DNA Real-Time PCR

The standard curve of real-time PCR was made with a DNA template amplified from CyHV-3-U DNA that was cloned into a TOPO 2.1 PCR vector (Invitrogen), as reported previously [8]. The copy number of CyHV-3 DNA was calculated from the equation derived from the standard curve, y = −1.267ln(x) + 37.202, where x is the Ct value. The PCR reaction was performed as previously described [8]. The primers used in real-time PCR were CyHV-3 86F, CyHV-3 163R, and Taqman probe CyHV-3 109P, as previously developed by Gilad et al. [4].

### 2.8. Statistical Analysis

All statistical analyses were performed using GraphPad Prism version 8.0.2 for Windows (GraphPad Prism 8.0.2, San Diego, CA, USA). Two-way ANOVA with multiple comparison tests was used to analyze virus titer, CyHV-3 genome copy numbers, and koi survival.

## 3. Results

### 3.1. HAL Reduces CyHV-3 Replication In Vitro

The HAL against CyHV-3 infection was evaluated after KF-1 cells were infected with ~500 PFU CyHV-3 per plate. The infected cells were exposed to HAL at 5 µM, 10 µM, and 20 µM for 2 h, respectively, following 1 h post-infection. The PFU was counted on day 10 post-infection. As shown in Figure 1A, nearly 90% inhibition of CyHV-3 plaque formation was observed in cells treated with 5 µM, 10 µM, and 20 µM for 2 h, which suggests the HAL has an EC90 against CyHV-3 around 5 µM. Around 50% inhibition was observed in cells treated with ACV at 10 µM or 20 µM for 2 h following 1 h post-infection (Figure 1B). In addition, infected cells treated for 6 h at 5 µM or 10 µM did not significantly enhance HAL antiviral activity except at 20 µM (Figure 1C). Cells treated with 20 µM HAL for 6 h reduced CyHV-3 PFU significantly compared to those treated with 5 µM and 10 µM, with a *p*-value at 0.04 and 0.009, respectively (Figure 1C). These results suggest that HAL at 5 µM is as effective as 20 µM against CyHV-3 replication within 2 h of treatment time, but 20 µM had a higher inhibition than 5–10 µM did when the treatment is 6 h.

### 3.2. HAL Treatment Effect Is Different with Treatment Time Against Different Viral Dose

The HAL treatment was further investigated with 10 µM HAL for 30 min, 60 min, 2 h, and 6 h, respectively, immediately after CyHV-3 absorption. As shown in Figure 2A, HAL treatment at 30, 60 min, 2 h, and 6 h all significantly reduced virus replication. There was no significant difference between 30 min vs. 60 min and 2 h vs. 6 h, but there were significant differences between the 60 min and 2 h or 6 h treatment, with *p* values at 0.012 and 0.0121, respectively. This suggests HAL treatment is time dependent. Since HAL at 5–20 µM has little dose effect against CyHV-3, HAL concentration between 0.5 µM and 5 µM was further evaluated. Immediately after a one-hour infection exposure of ~1000 FPU/plate, cells were treated with 0.5 µM, 1 µM, and 5 µM HAL for 2 h. As shown in Figure 2B, HAL at 0.5 µM, 1 µM, and 5 µM can block nearly 30%, 40%, and 49% plaque formation in vitro, respectively. This suggests HAL treatment is dose dependent. To assess the strength of HAL against CyHV-3 replication, HAL against different doses of CyHV-3 was evaluated at 5 µM. When the input dose is at ~1000 PFU per plate, HAL at 5 µM can block about 50% plaque formation, but can block around 90% of CyHV-3 infection when the input virus is ~500 PFU (Figure 2C). This result suggests that HAL has an EC50 around 5 µM against ~1000 PFU and an EC90 at about 5 µM against ~500 PFU.

### 3.3. HAL Has a Treatment Effect Within the First 5 Days Post-Infection

The window of treatment against CyHV-3 infection was evaluated on 1, 3, and 5 days post-infection with 20 µM HAL treatment. As shown in Figure 3A, close to 95% and 94% PFU reduction was seen when the treatment was applied on 1 and 3 dpi, and around 64% reduction was observed on 5 dpi, respectively. The treatment applied within 5 dpi was still statistically significant compared to the untreated control, with a *p*-value around 0.0004. When viral genome replication was measured, a significant inhibition against viral DNA replication can be seen with 20 µM HAL treatment for 2 h when the drug was given on 3 (*p* = 0.01) and 5 dpi (*p* = 0.049) (Figure 3B,C), while 10 µM HAL treatment for 1 h was no longer effective when the drug was given on 5 dpi (Figure 3C). These results indicate that HAL is only effective against CyHV-3 during the early phase of infection.

### 3.4. Cytotoxicity of HAL

To determine whether HAL is cytotoxic to KF-1 cells, the cell viability was evaluated after exposure to HAL at different concentrations for 24 h. Figure 4A shows viability results from KF-1 cells treated with HAL over a concentration of 1 to 100 µM for 24 h. No significant difference in viability was apparent between mock-treated and 24 h treated cells with up to 50 µM HAL in KF-1 cells. However, a statistically significant difference was noticeable between mock-treated and 100 µM HAL-treated KF-1 cells (*p* = 0.001). Since 10 µM and 20 µM HAL is non-toxic to KF-1 cells and can block about 90% of 1 × 10^3^ PFU CyHV-3/plate infection in vitro, these two concentrations were selected for the in vivo study. ACV cytotoxicity was measured similarly to HAL in KF1 cells. No apparent cytotoxicity was observed in KF-1 cells treated with ACV at 1–100 µM for 24 h (Figure 4B).

### 3.5. HAL Reduced Mortality in CyHV-3 Infected Younger Koi

CyHV-3 infection at 15–17 °C does not cause high mortality. In order to determine the drug protection effect against mortality associated with CyHV-3 infection, we found that a daily temperature increase mimics conditions related to the high mortality associated with CyHV-3 infection in koi [25]. To determine whether a HAL immersion bath treatment could reduce mortality in 3–4-month-old koi, four groups of koi were infected with ~1 × 10^3^ PFU/mL of CyHV-3 for 30 min in an immersion bath using the infection temperature model developed in a previous study [26]. Following CyHV-3 exposure, the tank temperature was increased at 2 °C per day from day 1 post-infection up to 23 °C on day 4 post-infection; the tank temperature was maintained at 23 °C for two days and then was decreased 2 °C per day until 15 °C (Figure 5A). Under these temperature changes, CyHV-3- infected koi at 3–4 months old will have ~80–90% mortality, as reported in our previous study [26]. From days 1 to 5 post-infection, 20 koi per group were treated daily in an immersion bath containing 10 µM HAL, 20 µM HAL, 20 µM ACV, or dilution media (DMSO) for 3–4 h, respectively. Koi were considered dead when floating without gill movement in the water or sunken to the bottom of the tank. Similarly to the previously reported death rate [26], the DMSO-treated group had ~80% mortality by day 15 post-infection, and mortality started as early as day 5 post-infection (Figure 5B). However, the 10 µM HAL-treated group had only about ~40% mortality by day 15 post-infection and mortality started two days later than that in the DMSO-treated group. The ACV-treated group had about 78% mortality by day 15 post-infection, which was slightly lower than that in the DMSO-treated group but not significantly different at the end of the experiment (Figure 5). ACV treatment resulted in a one-day delay in mortality compared with the DMSO-treated group. Interestingly, the 20 μM HAL-treated group had a similar outcome as those treated with ACV.

### 3.6. HAL Reduced CyHV-3 Reactivation from Temperature Stress

As we reported previously, CyHV-3 reactivation in latently infected koi can be induced by temperature stress [8]. Koi which have recovered from CyHV-3 infection will become latently infected with the virus. CyHV-3 becomes latent in a small percentage of B cells in recovered koi, which can only be detected by PCR using the total DNA of white blood cells (WBCs) [11,24]. During reactivation, CyHV-3 DNA increase in the peripheral white blood cells can be detected between day 1 and 10 after temperature stress. To determine whether HAL can reduce CyHV-3 reactivation from latency, CyHV-3 latently infected koi kept from a previous study, where were 25–30 cm (4–6 years old), were treated with HAL intramuscularly (IM) during temperature stress (Figure 6A). Four koi per group were injected IM with 100 µL of 20 µM harmaline or 100 µL of PBS (treatment control) daily for 5 days on day 1 after temperature stress, respectively. Another four control koi were not stressed or treated. Blood samples of treated or untreated koi were taken on 1, 3, 6, and 9 days after temperature stress (dps) (Figure 6A). Since CyHV-3 becomes latent in a small percentage of B cells, low levels of CyHV-3 DNAcould be detected by qPCR in the total WBC during latency in the untreated and unstressed control koi (Figure 6, black bar, cc). Following 24 h after temperature stress, there was a significant CyHV-3 DNA increase in both PBS and HAL-treated koi (Figure 6B). However, CyHV-3 DNA levels were significantly reduced on 3 and 6 dps in groups treated with HAL compared to those treated with PBS, with a *p*-value at 0.033 on 3 dps and 0.023 on 6 dps, respectively. By 9 dps, when the temperature is near normal, CyHV-3 genome copies in WBCs from stressed and treated koi were all back down to the baseline level seen in the untreated and unstressed koi. These results suggest that HAL can lower the CyHV-3 reactivation rate in latently infected koi.

## 4. Discussion

Koi are colored variants of carp and are valuable ornamental fish collected by many koi hobbyists throughout the world. CyHV-3 is the most pathogenic virus in koi and common carp, especially in younger koi [27,28,29]. Currently, no effective antiviral drugs have been developed against CyHV-3 infection in koi [30,31,32]. Anti-herpesvirus drugs, such as acyclovir (ACV), acyclovir monophosphate (ACV-MP), and Arthrospira platensis, have been investigated against CyHV-3 and were found to have limited effects against CyHV-3 infection in vitro and in vivo [33,34,35]. Our study also demonstrated that ACV has limited protection against CyHV-3 infection in vitro and in vivo (Figure 1B and Figure 5). Therefore, antiviral drugs or alternative medicines are needed to control CyHV-3 infection in koi. The analogous β-carboline alkaloids, such as harmaline (HAL) and harmine (HAR), possess a variety of biological properties that have inhibitory activity against acetylcholinesterase (AChE), monoamine oxidase (MAO), and myeloperoxidase (MPO). HAR has been shown to be neuroprotective, anticancer, antimicrobial, and antiviral [12,36]. HAR, isolated from the seeds of the medicinal plant *Peganum harmala* L., has been used for thousands of years in the Middle East and China. Recently, HAR derivatives have been reported to have antiviral activities against various viruses, such as HSV-1, HSV-2, Newcastle disease virus, dengue virus, and polio virus [19,20,21,37,38,39]. Here, we demonstrated that HAL, which is similar to HAR in structure, also has anti-viral activities against CyHV-3 in vitro and in vivo.

Both HAR and HAL can be found in various plants, such as Syrian rue and ayahuasca, which have been used in traditional medicine for centuries. Previous studies have shown HAR has antiviral properties against HSV-1 and HSV-2, HIV, and coronavirus [19,21,38]. Although HAL has not been tested directly against herpesvirus in vitro, it has a similar structure to HAR and has better bioavailability and solubility than HAR [13]; therefore, HAL was tested in this study. Our study found that HAL is more soluble than HAR and has anti-CyHV-3 activity at relatively low concentrations, such as 5 µM, 10 µM, and 20 µM, in vitro (Figure 1 and Figure 2). Those concentrations were all below the toxic concentration (Figure 4). Another interesting finding is that when KF-1 cells infected with ~500 PFU CyHV-3 were treated for 30 min, HAL at 5 µM had relatively higher antiviral activity than those treated with 10 µM or 20 µM (Appendix A). The difference between 5 µM and 10 µM is statistically significant, but it is not significant between 5 µM and 20 µM. This suggests that the treatment time and HAL concentration may have different effects during CyHV-3 replication. Another interesting observation is that there is no significant dose effect between 5 µM and 20 µM at 2 h treatment against CyHV-3 in vitro. However, HAL at lower concentrations, between 0.5 µM and 5 µM, did have dose effects in vitro (Figure 2B). Since HAL is less soluble at 50 µM, the dose effect above 20 µM was not investigated. These results suggest that targeting cellular responses against virus replication is a balancing act between the time and dose of the chemicals.

It has been reported that HSV-1 infection will trigger the production of reactive oxygen species (ROS) [40,41] and oxidative stress during the early stage of replication [42]. ROS production increases are also associated with other virus infections, such as SARS-CoV-2, influenza, respiratory syncytial virus, and rhinoviruses, and are beneficial for viral replication [43]. The mode of action of HAR against HSV infection was found to be through the downregulation of cellular NF-kB and MAPK pathways induced by oxidative stress [20]. Therefore, antioxidants were shown to have antiviral activities against various viral infections [44]. It is reasonable to speculate that HAL will also have an antioxidant function and interfere with oxidative stress during CyHV-3 replication [14]. Here, we found that the effective treatment time is between 30 min and 2 h within the first 5 days post-infection: longer treatment duration seems to have no significant effect against CyHV-3 replication (Figure 1C). It suggests that HAL may interfere with viral replication by decreasing oxidative stress during the early phase of the infection.

Mortality associated with CyHV-3 infection in koi is age- and temperature-dependent [26]. Koi, at younger ages, are more susceptible to CyHV-3 infection [45]. Our previous study found that the infection of under 6-month-old koi with CyHV-3 at about 1 × 10^3^ PFU/mL via immersion did not cause any mortality at 15 °C [26]. However, infection of the same age groups of koi with the same infection dose could lead to 80–90% mortality within 10 days post-infection by increasing water temperature daily at 2 °C per day from 15 °C to 23 °C [26]. In this study, we infected a similar age group of koi and exposed them to a similar heat stress as reported previously [26]; we found 10 µM HAL-treated koi had a significantly higher survival rate than the rate in the control group (MDSO-treated group) (Figure 5B). No significant protection was observed in koi treated with ACV, which is in agreement with our previous research [26]. Another interesting finding is that the 20 µM HAL-treated group had similar survival as those treated with the 20 µM ACV-treated group. This could be a similar effect seen in Appendix A. The HAL antiviral treatment effect is complicated by treatment time and treatment concentration at different temperatures. Since HAL treatment was applied during temperature increases, higher concentrations of HAL may also have adverse effects on the hosts. More studies will be needed to determine the optimal concentration, treatment time, and temperature. This could mean that HAL bioactivity is limited to 10 µM, and an extra amount or longer treatment time at a higher temperature may have a negative effect on the host. Nevertheless, this study demonstrated HAL at 10 µM can decrease KHV replication and reduce mortality associated with the infection.

Our previous studies have demonstrated that CyHV-3 becomes latent in WBCs, especially the IgM^+^ positive B cells [24]. CyHV-3 reactivation can be induced by the daily temperature increasing at 2 °C per day from 15 °C to 23 °C and then maintained at 23 °C for 3–5 days [8]. A two- to three-fold CyHV-3 genome copy increase in WBCs can be detected between 3 and 9 dps [11]. Similarly, a 2–3-fold CyHV-3 DNA increase in WBCs was detected between 1 and 6 dps in koi treated with PBS (Figure 6B). However, CyHV-3 DNA increase was only observed between 1 and 3 dps in those HAL-treated koi. By 6 dps, HAL-treated koi had a significantly lower CyHV-3 DNA in total WBCs collected from 2 mL blood. The HAL treatment effect was noticeable on 3 and 6 dps. This suggests that HAL treatment can lower CyHV-3 reactivation if CyHV-3 latently infected koi are treated with HAL via IM. This treatment could be used in koi shipping to prevent CyHV-3 from reactivating due to shipping stress.

## Figures and Tables

**Figure 1 viruses-17-00687-f001:**
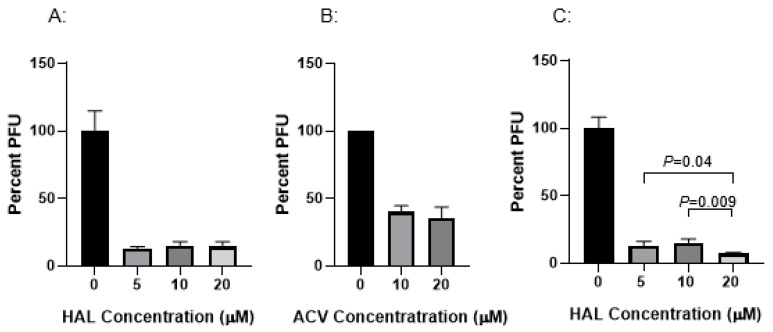
Percentage of CyHV-3 plaque formation units (PFUs) in the presence of HAL and ACV at different treatment concentrations and treatment times. Plates seeded with KF-1 cells were treated with HAL (**A**,**C**) and ACV (**B**) in triplicates following an hour absorption with ~500 PFU of CyHV-3-U/plate. Each set was treated with HAL at 5 µM, 10 µM, and 20 µM or ACV at 10 µM and 20 µM for 2 h (**A**,**B**) or 6 h (**C**), respectively. After the treatment, each plate was washed once with PBS and covered with 3% methylcellulose overlay media. Plaque formation units (PFUs) were counted on day 10 post-infection. A significant statistical difference in treatment concentration is marked with a *p*-value calculated by two-way ANOVA with a Bonferroni post-test.

**Figure 2 viruses-17-00687-f002:**
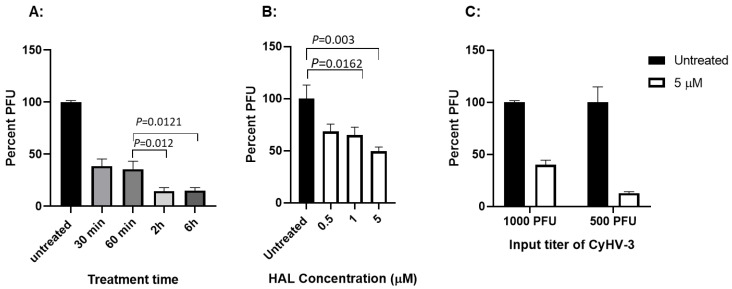
Percentage of CyHV-3 PFU in the presence of HAL treatment for different times and against different viral doses. (**A**): Percentage of CyHV-3 PFU in the presence of HAL for different times. KF-1 cells were infected as described in Figure 1 and then treated with 10 µM for 30 min, 60 min, 2 h, and 6 h, respectively, immediately after one hour of virus absorption. After the treatment, each plate was washed once with PBS and covered with 3% methylcellulose overlay media. (**B**,**C**): Percentage of CyHV-3 PFU following 0.5–5 µM HAL treatment (**B**) or cells infected with different amounts of the virus (**C**). KF-1 cells were infected with ~1000 PFU or ~500 PFU CyHV-3 per plate and then treated with 0.5–5 µM for 2 h, immediately after one hour of virus absorption. Each treatment was repeated in triplicates (n = 3). The PFU was counted on day 10 post-infection. A significant statistical difference between different treatment times is marked with a *p*-value calculated by two-way ANOVA with a Bonferroni post-test.

**Figure 3 viruses-17-00687-f003:**
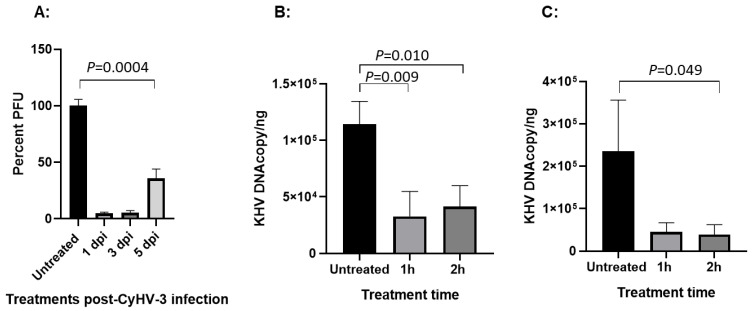
Percentage of PFU and viral genome copy number in CyHV-3 infected KF-1 cells treated with HAL on different days post-infection (dpi). (**A**): Percentage of CyHV-3 PFU in the cells treated with HAL at 20 µM for 2 h on 1, 3, and 5 dpi after the cells were infected with ~300 PFU/plate. (**B**,**C**): qPCR of CyHV-3 in KF-1 cells treated with HAL on different dpi. KF-1 cells were infected as above and then treated with HAL at 20 µM for 1 h or 2 h on 3 dpi (**B**) and 5 dpi (**C**), respectively. The number of CyHV-3 genome copies was estimated by qPCR using total DNA isolated from infected cells on day 10 post-infection, as described previously [11]. A significant statistical difference between different treatments is marked with a *p*-value calculated by two-way ANOVA with a Bonferroni post-test.

**Figure 4 viruses-17-00687-f004:**
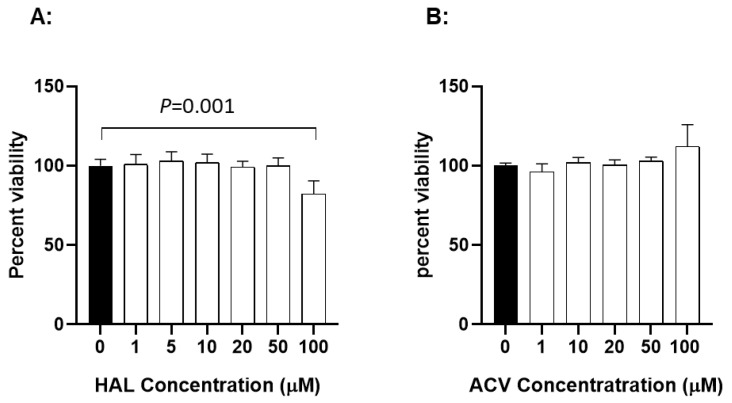
The cytotoxicity of HAL (**A**) and ACV (**B**) in vitro. KF-1 cells were incubated with the indicated concentration of HAL or ACV for 24 h. The treatment was then removed and the cells were washed once with PBS and further incubated for 24 h in tissue culture media with 5% FBS and antibiotics, as described in the materials and methods. Cell viability was evaluated with an XTT cell viability kit III (PromoKine) and expressed as a percentage of the mock-treated control (n = 3). A significant statistical difference between the mock-treated control (0) and 100 µM HAL is marked above the line with a *p*-value calculated by two-way ANOVA with a Bonferroni post-test.

**Figure 5 viruses-17-00687-f005:**
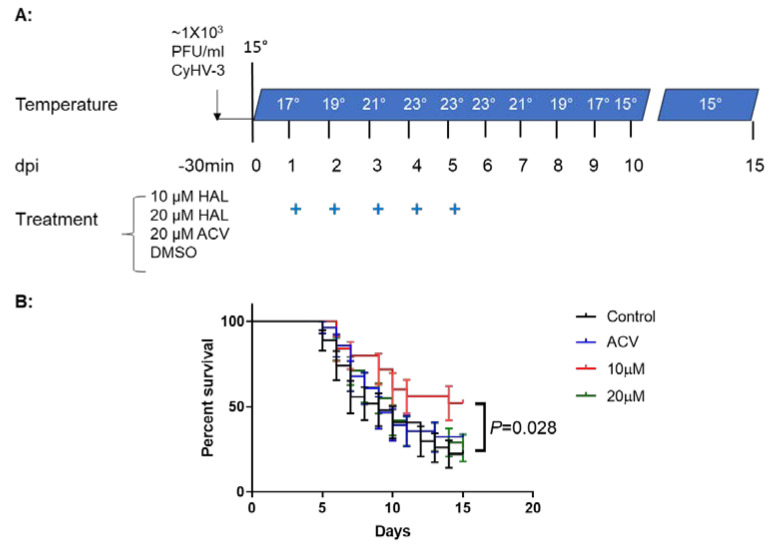
Schematic of the infection-treatment protocol and survival of koi infected with CyHV-3 in different treatment groups. (**A**): 3–4-month-old koi were infected with 1 × 10^3^ PFU/mL for 30 min, followed by water temperature change on different dpi and treatment for five consecutive days with HAL, ACV, or dilution medium (DMSO) as indicated. The “+” stands for days that koi were treated with 10 µM HAL, 20 µM HAL, 20 µM ACV, or DMSO (control), respectively. (**B**): The percentage survival of koi infected with CyHV-3 in four different treatment tanks with 10 µM HAL, 20 µM HAL, 20 µM ACV, or DMSO (control), respectively. A significant statistical difference between the DMSO-treated and 10 µM HAL-treated groups is marked with a *p*-value calculated using a two-way ANOVA.

**Figure 6 viruses-17-00687-f006:**
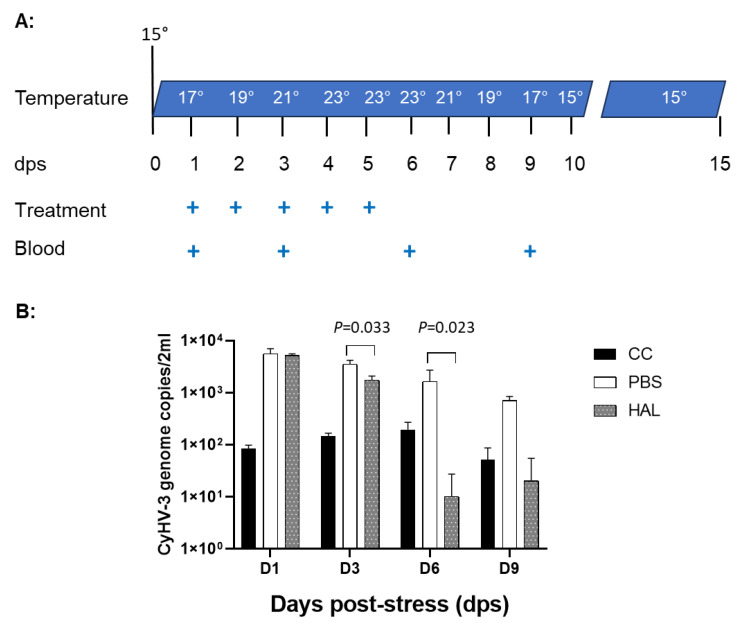
Schematic of treatment protocol during temperature stress and CyHV-3 genome copy numbers in peripheral white blood cells. (**A**). The water temperature changes as indicated on different days after temperature stress (dps). The “+” stands for dps when treatment was given or blood was sampled. The treatment was given IM at 100 µL of 20 µM HAL or 100 µL of PBS. (**B**): CyHV-3 genome copy numbers were estimated by qPCR using total DNA of WBCs from 2 mL blood collected from each koi on 1, 3, 6, and 9 dps. The number represents the average CyHV-3 genome copies in WBCs from 2 mL of blood per group. The “cc” stands for koi that were not stressed or treated. A significant statistical difference between the HAL-treated and PBS-treated groups on 3 and 6 dps is marked with a *p*-value calculated using a two-way ANOVA with a Bonferroni post-test.

## Data Availability

The original contributions presented in this study are included in the article. Further inquiries can be directed to the corresponding author.

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
