# Peer review of "Investigation of β-Carboline Alkaloid Harmaline Against Cyvirus cyprinidallo3 Infection In Vitro and In Vivo"

_viruses, 2025, doi:10.3390/v17050687_

Round 1

Reviewer 1 Report

Comments and Suggestions for Authors

In this manuscript , Manes et al investigated the antiviral activity of beta-carboline harmaline (HAL) against Cyprinidallo herpesvirus 3 (formally named Cyprinid herpesvirus 3 or Koi herpesvirus). Acyclovir was used in this MS  as a reference antiviral molecule. Toxicity of HAL was tested in vitro at concentrations between 1 and 100 micromolar.  The author tested the antiviral activity of the drug in a range of concentrations ranging from 5 to 20 micromolar with no evidence of dose-effect relationship (Fig. 1-3). The antiviral effect was also tested in vivo at 10 and 20 micromolar. The author claimed that HAL treatment at 10 micromolar for 5 days increased the survival rate by 60% in inoculated juvenile fish. Finally, the author claimed that their treatment can reduce the rate of reactivation occurring in latently infected fish submitted to a thermal stress.

Major comments

1/ The manuscript contains a large number of typographic or nomenclature errors. For example, the author should respect the ICTV nomenclature.

2/ It is surprising that HAL was not tested in a range of concentrations allowing the observation of a dose effect relationship. Of note the figures related to activity contains very few data with some of them being redundant between figures (Fig. 1C and Fig. 2).

3/ Basic studies on antiviral activity encompasses dose-effect on viral production based on titration of virions. This basic experiment is absent here

4/ Fig. 4 is the source of important and serious critics.  The design of the experiment is unusual and raised concerns. Why submitting naïve subjects to a thermal stress for a simple CyHV-3 challenge? More surprisingly, the thermal regime imposed to fish started and ended with temperatures which are known to be non-permissive to CyHV-3 disease. Concerning the analysis of the data, the authors claimed a 60% increase of survival in the 10 micromol group (which obviously generated the largest heterogeneity of data) while the group treated with the double dose had no protective effect. Together these critics invalidate the significance of this experiment, raised questions about the relevance of the experiment design  and so the bioethics foundation to perform such experiment.

5/ Fig. 6. The authors claimed that they investigated the rate of reactivation. To determine the rate of reaction, one must identify the viral charge baseline in individual latently infected subject than measured the viral charge after the reactivation stress with or without treatment. The experiment performed by the authors is different and indicated only that the treatment affect the global mean viral charge.

Comments on the Quality of English Language

See general comments

Author Response

In this manuscript , Manes et al investigated the antiviral activity of beta-carboline harmaline (HAL) against Cyprinidallo herpesvirus 3 (formally named Cyprinid herpesvirus 3 or Koi herpesvirus). Acyclovir was used in this MS  as a reference antiviral molecule. Toxicity of HAL was tested in vitro at concentrations between 1 and 100 micromolar.  The author tested the antiviral activity of the drug in a range of concentrations ranging from 5 to 20 micromolar with no evidence of dose-effect relationship (Fig. 1-3). The antiviral effect was also tested in vivo at 10 and 20 micromolar. The author claimed that HAL treatment at 10 micromolar for 5 days increased the survival rate by 60% in inoculated juvenile fish. Finally, the author claimed that their treatment can reduce the rate of reactivation occurring in latently infected fish submitted to a thermal stress.

RESPONSE: Regarding the dose effect against CyHV-3, the range of concentration between 5µM and 20µM did not show a dose effect against CyHV-3 when cells were treated for 1 or 2h. We further investigated the dose effect between 0.5 µM and 5 µM. As shown in the new Fig. 2B, HAL did have a dose effect between 0.5 µM and 5µM. In addition, 20µM HAL at 6h treatment did have a significant inhibition against CyHV-3 replication compared to 5µM and 10 µM treatment. HAL does have a dose effect depending on the treatment time and concentration.

Major comments

  1. The manuscript contains a large number of typographic or nomenclature errors. For example, the author should respect the ICTV nomenclature.

RESPONSE: The typographic and nomenclature errors have been corrected in our best efforts.

  1. It is surprising that HAL was not tested in a range of concentrations allowing the observation of a dose effect relationship. Of note the figures related to activity contains very few data with some of them being redundant between figures (Fig. 1C and Fig. 2).

RESPONSE: The common antiviral treatment concentration is normally studied between 10 µM and 20 µM. Therefore, HAL concentrations ranging from 5µM to 20µM were investigated first in this study. Since 5 µM has a similar effect with 10µM or 20µM, the HAL dose effect below 5µM is now further investigated. As shown in new Fig. 2B, HAL between 0.5µM and 5µM did have a dose effect against CyHV-3 in vitro. The protection of HAL at 5µM against CyHV-3 also depends on the input dose of viruses (new Fig. 2C). In addition, 20µM HAL at 6h treatment did have a significant inhibition against CyHV-3 replication compared to 5µM and 10µM treatment (Fig. 1C). HAL is less soluble at 50µM; thus, the dose-effect at higher concentrations is not investigated.

  1. Basic studies on antiviral activity encompasses dose-effect on viral production based on titration of virions. This basic experiment is absent here

RESPONSE: The HAL antiviral effect is measured by its ability to block CyHV-3 plaque formation, which correlates to infectious virion production. The input dose at 500-1000 PFU units corresponds to about 500-1000 infectious virions. When the cells are infected with ~ 1000 PFU virus, HAL at 5µM can only block around 50% of the virus replication, but it can block over 90% of plaque formation when the input dose is ~500 PFU per plate (Fig. 2C).

  1. Fig. 4 is the source of important and serious critics. The design of the experiment is unusual and raised concerns. Why submitting naïve subjects to a thermal stress for a simple CyHV-3 challenge? More surprisingly, the thermal regime imposed to fish started and ended with temperatures which are known to be non-permissive to CyHV-3 disease. Concerning the analysis of the data, the authors claimed a 60% increase of survival in the 10 micromol group (which obviously generated the largest heterogeneity of data) while the group treated with the double dose had no protective effect. Together these critics invalidate the significance of this experiment, raised questions about the relevance of the experiment design and so the bioethics foundation to perform such experiment.

RESPONSE: Do you mean Fig. 5? Regarding why submitting naïve subjects to thermal stress for a simple CyHV-3 challenge? CyHV-3 infection at 15-17°C does not cause much mortality in Koi under 6 months old when they were infected with ~1x103 PFU/ml. In order to assess the drug protection effect against mortality associated with CyHV-3 infection, we found that changing temperature daily, starting from a non-permissive temperature of 15°C to a permissive temperature of 23°C could lead to higher mortality in Koi from CyHV-3 infection. Our recent publication reported this finding (Matsuoka S et al. 2023). Here, we used the same infection temperature model to produce the mortality associated with CyHV-3 infection to evaluate HAL antiviral activities.

Regarding 20µM treatment with less effect against CyHV-3 infection in koi, this could result from the different treatment times at different treatment temperatures. Since the treatment was performed during temperature increases (days 1 to 5), higher HAL concentrations may have adverse effects on the host. Lower concentrations may work better during temperature rise. The mechanism behind the difference is another subject of research that is beyond the scope of this study.

  1. Fig. 6. The authors claimed that they investigated the rate of reactivation. To determine the rate of reaction, one must identify the viral charge baseline in individual latently infected subject than measured the viral charge after the reactivation stress with or without treatment. The experiment performed by the authors is different and indicated only that the treatment affect the global mean viral charge.

RESPONSE: Koi recovered from CyHV-3 infection will all become latently infected with the virus. All koi used in the reactivation study were survivors of CyHV-3 infection in the past and kept in the John L. Fryer Aquatic Animal Health Lab at Oregon State University for the past 3-5 years. CyHV-3 becomes latent in a small percentage of B cells in recovered Koi (Reed AN et al. JV 2014). During latency, only low levels of CyHV-3 DNA can be detected by real-time PCR or nested PCR in total WBC from 2ml of blood (Reed AN et al. JV 2014).  CyHV-3 can be reactivated in latently infected koi following temperature stress (Eide K. et al. 2011). An increase of CyHV-3 DNA copies could be detected in the WBC during CyHV-3 reactivation from latency following temperature stress (Lin L. et al. 2017. Virus Research). Therefore, to determine if HAL treatment has any effect against CyHV-3 reactivation, latently infected koi were stressed with a similar temperature change regimen as reported previously (Eide K. et al. 2011; Lin L. et al. 2017), and CyHV-3 DNA level in the total WBC was monitored under different treatments. The baseline KHV DNA did not change between 1- and 9-days post-stress (dps) in those untreated control koi. Among treated and stressed koi, total CyHV-3 DNA immediately went up above the baseline of CyHV-3 within 1 dps in both HAL-treated and PBS-mocked treated groups. However, CyHV-3 DNA level decreased significantly between 3 and 6 dps in the HAL-treated group. This indicates that HAL treatment reduced CyHV-3 reactivation in those latently infected Koi induced by temperature stress. By 9 dps, when the temperature was near normal, CyHV-3 DNA levels in all treated and untreated Koi returned to the baseline. Additional rationale for our approaches has now been included in this section.

Reviewer 2 Report

Comments and Suggestions for Authors

          Harmine (HAR) and Harmaline (HAL) are β-carboline alkaloids found in the medicinal plant Peganum harmala with antiviral activities against many viruses. Here, the authors demonstrated that HAL, which is similar to HAR in structure, has anti-viral activities against CyHV-3 in vitro and in vivo; immersion treatment with 10µM HAL for 3-4h daily within the first 5 days 
post-infection can increase the survival of fry by 60%, and IM injection of HAL at 20µM can reduce the rate of KHV reactivation induced by heat stress in latently infected koi. These results are very interesting and encouraging, and should pave the way for development anti-CyHV3 strategy that is urgently needed due to the lack of commercial vaccine or medicine.

            minor concerns are listed here:

  1.  To determine whether HAL has anti-viral effect against CyHV-3 replication in  koi fin cells (KF-1), the EC50 value should be calculated to evaluate or compare antivial effects among difefrent chemicals. The authors described a lot of results of various infection time/treatment dosage,  but lack of a simple index to validate the antiviral effect.
  2.  CyHV-3 DNA real-time PCR  and  CyHV-3 plaque reduction assay were used in the study. Can the authors compare the plaque assay data with qPCR result to evaluate the accuracy of qPCR in quantitating virus?
  3. In 3.4, it 's not clear how do the authors prepare CyHV-3 latently infected Koi? Is it prepared by laboratory challenge and just recovered fish from an infection experiment? 

Author Response

Harmine (HAR) and Harmaline (HAL) are β-carboline alkaloids found in the medicinal plant Peganum harmala with antiviral activities against many viruses. Here, the authors demonstrated that HAL, which is similar to HAR in structure, has anti-viral activities against CyHV-3 in vitro and in vivo; immersion treatment with 10µM HAL for 3-4h daily within the first 5 days 
post-infection can increase the survival of fry by 60%, and IM injection of HAL at 20µM can reduce the rate of KHV reactivation induced by heat stress in latently infected koi. These results are very interesting and encouraging, and should pave the way for development anti-CyHV3 strategy that is urgently needed due to the lack of commercial vaccine or medicine.

      minor concerns are listed here:

  1. To determine whether HAL has anti-viral effect against CyHV-3 replication in  koi fin cells (KF-1), the EC50 value should be calculated to evaluate or compare antivial effects among difefrent chemicals. The authors described a lot of results of various infection time/treatment dosage,  but lack of a simple index to validate the antiviral effect.

RESPONSE: The revised manuscript now includes the EC50 and EC90 for HAL against different infection doses, which is shown in the new Fig. 2C. HAL has an EC50 around 5µM if the input virus is around ~1000 PFU/plate and an EC90 around 5µM if the input virus is around ~500 PFU/plate.

 CyHV-3 DNA real-time PCR  and  CyHV-3 plaque reduction assay were used in the study. Can the authors compare the plaque assay data with qPCR result to evaluate the accuracy of qPCR in quantitating virus?

RESPONSE: A new Figure 3A in PFU reduction has been included in the revised manuscript.

  1. In 3.4, it 's not clear how do the authors prepare CyHV-3 latently infected Koi? Is it prepared by laboratory challenge and just recovered fish from an infection experiment? 

RESPONSE: The latently infected koi were fish recovered from CyHV-3 infection and kept in the John L. Fryer Aquatic Animal Health Lab at Oregon State University for 3-5 years.

Reviewer 3 Report

Comments and Suggestions for Authors

The manuscript “Investigation of b-Carboline Alkaloid Harmaline against Cyprinid herpesvirus 3 infection in vitro and in vivo” by Manes et al. is an article about the treatment of CyHV-3 with Harmaline. The authors make a clear introduction into the topic and explain the reason for conducting this study well. The presented study design is suitable and well conducted. The authors found HAL to be effective against CyHV-3. However, additional information are missing. What is the mode of action of HAL against CyHV-3? Does ROS occur after infection and is the HAL an effective antioxidant in carp?

Some specific comments:

  • According to the new nomenclature it is Cyvirus cyprinidallo
  • In deed there are effective vaccines against CyHV-3, even though they are not produced.
  • In science SI-system is commonly used. Please avoid archaic units like “inch” and use metre or centimetre.
  • How do you explain the difference of 10µM and 20µM HAL in Fig. 5?

Author Response

The manuscript “Investigation of b-Carboline Alkaloid Harmaline against Cyprinid herpesvirus 3 infection in vitro and in vivo” by Manes et al. is an article about the treatment of CyHV-3 with Harmaline. The authors make a clear introduction into the topic and explain the reason for conducting this study well. The presented study design is suitable and well conducted. The authors found HAL to be effective against CyHV-3. However, additional information are missing. What is the mode of action of HAL against CyHV-3? Does ROS occur after infection and is the HAL an effective antioxidant in carp?

RESPONSE: There are production of reactive oxygen species (ROS) [40, 41] and oxidative stress during the early stage of HSV-1 replication [42]. ROS production increases are also associated with other virus infections, such as SARS-CoV-2, influenza, respiratory syncytial virus, and rhinoviruses, and are beneficial for viral replication (44). Therefore, antioxidants were shown to have antiviral activities against various viral infections [43]. The mode of action of HAR against HSV infection was reported to be through the downregulation of cellular NF-kB and MAPK pathways induced by oxidative stress (20). We did not measure ROS during the treatment of HAL, which is beyond the scope of this study. It is reasonable to speculate that HAL has similar antioxidant functions and interferes with oxidative stress during viral replication [21]. We hypothesized that the HAL has a mechanism of action similar to HAR, which can block ROS production during CyHV-3 infection, thus reducing its replication.

Some specific comments:

1. According to the new nomenclature it is Cyvirus cyprinidallo

RESPONSE: Correction has been made as suggested.

2. In deed there are effective vaccines against CyHV-3, even though they are not produced.

RESPONSE: Agree. However, the vaccine cannot prevent herpesvirus reactivation. Therefore, treatment options are needed to limit the mortality associated with CyHV-3 infection or reactivation.

3. In science SI-system is commonly used. Please avoid archaic units like “inch” and use metre or centimetre.

RESPONSE: Correction has been made as suggested.

4. How do you explain the difference of 10µM and 20µM HAL in Fig. 5?

RESPONSE: As mentioned above in response to reviewer one comment 4, the difference between 10µM and 20µM HAL could come from the treatment time and temperature changes during treatment. 20µM HAL for 3h treatment during temperature increases may have a negative effect on the host defense. As Sup Fig 1 shows, there is an inverse correlation in treatment against CyHV-3 during certain treatment times. The lower dose seems to work better than the higher dose dd in the shorter treatment. It is a balancing act of time, dose, and temperature when therapeutic chemicals target cellular responses during virus infection.